# Cardiovascular Implications of Sleep Bruxism—A Systematic Review with Narrative Summary and Future Perspectives

**DOI:** 10.3390/jcm10112245

**Published:** 2021-05-21

**Authors:** Monika Michalek-Zrabkowska, Helena Martynowicz, Mieszko Wieckiewicz, Joanna Smardz, Rafal Poreba, Grzegorz Mazur

**Affiliations:** 1Department of Internal Medicine, Occupational Diseases, Hypertension and Clinical Oncology, Wroclaw Medical University, 213 Borowska St., 50-556 Wroclaw, Poland; monika.michalek@student.umed.wroc.pl (M.M.-Z.); rafal.poreba@umed.wroc.pl (R.P.); grzegorz.mazur@umed.wroc.pl (G.M.); 2Department of Experimental Dentistry, Wroclaw Medical University, 26 Krakowska St., 50-425 Wroclaw, Poland; mieszko.wieckiewicz@umed.wroc.pl (M.W.); joanna.smardz@umed.wroc.pl (J.S.)

**Keywords:** sleep bruxism, RMMA, rhythmic masticatory muscle activity, cardiovascular risk, cardiovascular disease, arterial hypertension

## Abstract

Sleep bruxism is a common sleep-related behavior characterized as repetitive masticatory muscle activity. Genetic vulnerability to stress and anxiety is considered a basal component in the pathogenesis of bruxism events. Dysfunction of the autonomic nervous system related with an arousal during sleep is considered an underlying cause of the cardiovascular implications of sleep bruxism. Increased cardiovascular risk was previously linked with sleep conditions: for example, obstructive sleep apnea and insomnia, and sleep bruxism. The aim of present systematic review was to evaluate the current arguments on the relationship between sleep bruxism and cardiovascular diseases according to the Preferred Reporting Items for Systematic Reviews and Meta-Analyses (PRISMA). We have reviewed the Embase, PubMed (Medline) and Scopus databases to identify applicable articles (1994–2021). A total of 127 records in English language were identified, then after screening and exclusion of nonrelevant records, 19 full-text articles were evaluated. Finally, we included 12 studies for synthesis. Due to the heterogeneity of the compared studies, only a qualitative comparison and narrative summary were performed. In the majority of studies, increased sympathetic activity was successfully established to escalate heart rate variability, the inflammatory process, oxidative stress, endothelial remodeling and hormonal disturbances, leading to hypertension and other cardiovascular complications.

## 1. Introduction

One of the major topics investigated in the field of sleep conditions is sleep bruxism (SB). The widely adopted definition of bruxism constitutes that sleep bruxism is a rhythmic (phasic) or non-rhythmic (tonic) masticatory muscle activity during sleep and it is not a movement disorder or a sleep disorder in otherwise healthy subjects [1]. Clinical symptoms of SB are classified in the 3rd edition of International Classification of Sleep Disorders and involve regular or frequent bruxism events during sleep and the presence of abnormal tooth wear or incidents of jaw muscle pain or fatigue [2]. Most of the theories of bruxism etiology are focused on genetic and psychologic vulnerability to stress or anxiety [3], the role of neurotransmitters serotonin, dopamine, gamma aminobutyric acid (GABA) and noradrenaline [4], autonomic nervous system modulation [5] and exposure to exogenous risk factors, e.g., tobacco, alcohol or drugs and comorbidities [6]. The prevalence of sleep bruxism in the adult population is estimated to be about 8% [7,8] up to 13% [9]; however, it varies depending on age group. Widely accepted and used diagnostic criteria for sleep bruxism were proposed by the American Academy of Sleep Medicine (AASM) [10] and international consensus by Lobbezoo et al. [11], but polysomnography remains the gold standard [12].

Although sleep bruxism is considered a sleep behavior rather than sleep disorder, previous studies have emphasized its coexistence with sleep-related disorders, for example, obstructive sleep apnea. Most SB events are preceded by brain and cardiovascular activity. Thus, there are some questions that naturally arise from the link between SB and OSA. Cardiovascular implications of both sleep bruxism and sleep-related disorders are widely investigated. Furthermore, a considerable body of literature showed the association between sleep-related disorders and cardiovascular complications. For example, studies have provided evidence for the relationship between obstructive sleep apnea and stroke [13,14], resistant hypertension [15,16], cardiac arrythmias [17] and diabetes [18]. As has also been previously reported, insomnia and sleep deprivation are associated with substantial impairments, e.g., heart failure [19] and hypertension [20]. Primary and secondary restless leg syndrome (RLS) was also discussed as associated with increased risk of cardiovascular disease (CVD). Van Den Eeden et al. [21] demonstrated that primary RLS was linked with increased risk of hypertension, whereas secondary RLS was associated with coronary artery disease (CAD) and hypertension.

This review investigates a relatively new area which has emerged from the pathophysiology of sleep bruxism, establishing that increased sympathetic activity is a core of the causal chain with an initial increase in heart rate and microarousal accompanied by rhythmic masticatory muscle activity (RMMA). The key contribution of autonomic-cardiac network hyperreactivity can be described as homeostasis maintenance in response to cortical activation and an arousal [5]. The increased sympathetic activity may provide significant cardiovascular implications and it is considered as the main subject of this review.

## 2. Objectives—The Aim of the Systematic Review

The main objective of the present systematic review was to investigate the current arguments on the relationship between sleep bruxism and cardiovascular diseases according to the Preferred Reporting Items for Systematic Reviews and Meta-Analyses (PRISMA). We focused on the critical revision of evidence for cardiovascular risk and increased sympathetic activity in available original articles in the field of sleep bruxism. The authors also intended to evaluate the strengths and weaknesses of reviewed reports and, therefore, to develop a new approach to sleep bruxism consequences.

Our PICO question was as follows: Are patients with diagnosed sleep bruxism (P, I) at increased cardiovascular risk (O) compared with non-bruxing patients (C)?

## 3. Methods

This systematic review’s authors followed the Preferred Reporting Items for Systematic Reviews and Meta-Analysis (PRISMA) Checklist [22,23]. Only studies involving human adult sleep bruxers published in English language were included. The literature screening procedure was conducted with PRISMA guidelines; details are presented in the PRISMA flowchart in Figure 1. For the current systematic review, the search strategy included search terms as follows: sleep bruxism and (cardiovascular or sympathetic activity). The literature review was performed by three authors (M.M.-Z., H.M. and M.W.) who reviewed the Embase, PubMed (MEDLINE) and Scopus databases (access date 15 April 2021). A total number of 127 reports were found. Search results estimated 22 records in the Scopus database, 76 in Embase and 29 results in PubMed. Subsequently, duplicates were removed, and thorough screening of the titles and abstracts of the remaining articles was performed. The remaining 19 full-text records were analyzed for eligibility by two authors (M.M.-Z. and H.M.) independently. Finally, 12 manuscripts were included in the synthesis (see Table 1).

We have evaluated the quality of the evidence according to the GRADE system (Grading of Recommendations Assessment, Development and Evaluation) [37]. Results of the assessment are presented in Table 2. Most of the studies included in our systematic review (*n* = 10 from 12) demonstrated a statistically significant relationship between sleep bruxism and cardiovascular implications; others also revealed an association, but without statistical power due to poor analysis.

The most common quality of evidence score was low; 11 out of 12 studies were observational studies/case–controls. In 4 studies, we upgraded the result due to the large effect of the presented findings. On the other hand, a common cause of decreased grade was imprecision. One of the studies was designed as a randomized control trial [28].

## 4. Results

Most of the revised papers investigated bruxism events during sleep with polysomnography [24,25,26,29,32,33,34,35], three studies collected data using polygraphy devices [27,28,30] and one employed a sleep bruxism monitoring system composed of two accelerometers, an infrared camera, electroencephalography, electromyography and electrocardiography devices [31].

The investigated papers differed in terms of explored domain. Through analysis, we identified publications on autonomic function in bruxers [25,27,31]. Some studies focused on blood pressure in the context of sleep bruxism [29,30,32,35]; others evaluated changes in heart rate (heart rate variability, HRV) [24,26,28,33] in sleep bruxers. One study determined inflammatory markers and hormonal disturbances as cardiovascular risk factors [34].

Together, the findings of the presented studies confirm the association between sleep bruxism and cardiovascular response. The results of the research carried out by Sjöholm et al. [25] suggested that sympathetic vasoconstrictor function examined with cardiovascular reflex tests was disturbed in bruxers; however, tests were performed in waking state in relatively small sample size. Huynh et al. [27] separated three groups of patients: moderate to high SB, low SB and control. Then, a correlation analysis between sleep, RMMA events and HRV was performed. The authors concluded that an increase in sympathetic activity precedes SB onset in moderate to severe sleep bruxers [27]. Results of a recent study by Nukazawa et al. [31] demonstrated two things. First, they confirmed the association between autonomic nervous system activity and sleep bruxism events. Second, sympathetic nerve (SN) activity was correlated with SB event length, whereas parasympathetic nerve (PSN) activity was correlated with muscle activity (% maximum voluntary contraction). It is worth discussing the interesting fact that 93.3% of SB events (*p* < 0.01) demonstrated a pattern where SN activation was followed by beginning of SB activity and finished with PSN tone [31].

The literature pertaining to blood pressure surges in sleep bruxism strongly suggests this correlation. As has been previously reported by Nashed et al. [29], RMMA events were associated with blood pressure fluctuations. Moreover, the study concluded that arousals and body movements accompanying SB events can impact BP changes. BP recordings were based on beat-to-beat measures using finger cuffs [29]. The publication by Martynowicz et al. [32] presented some information about the background of renalase involvement in hypertension pathogenesis and indicated that there is an association between renalase concentration and sleep bruxism severity. The next study by Martynowicz et al. [30] was designed to evaluate the intensity of SB in a hypertensive group of patients compared with normotensives. Hypertension was found as an independent risk factor for increased bruxism episode index (BEI). A recent publication by Michalek-Zrabkowska et al. [35] revealed that systolic blood pressure variability during nighttime was significantly increased in severe bruxers (with BEI > 4/h) compared to individuals with BEI ≤ 4/h.

According to heart rate variability, seminal contributions have been made by Okeson et al., who aimed to collect a normative data on sleep bruxism. Results of this study revealed a cardiovascular response to bruxing event, with an increase in heart rate of 16.6% on average [24]. Zhong et al. [33] examined patients with polysomnography and then extracted three types of movements associated with heart rate increase: RMMA, RMMA+ limb movements and separate limb movements. Data of increased heart rate associated with RMMA events suggested autonomic activation to cortical arousal forasmuch as increased HR in response to different types of movements and RMMA related to respiratory events. Another interesting finding was revealed by Kato et al. [26] The study reported quantitative analysis of sequential changes observed in cortical electroencephalographic activity and sympathetic tone related to arousal and RMMA. A significant increase in heart rate reflecting autonomic-cardiac activation was observed in SB patients compared to the control group. Huynh et al. [28] employed an experimental randomized controlled study methodology which prescribes the use of two sympatholytic medications—propranolol and clonidine in sleep bruxers. Both medications decreased sympathetic tone, whereas only clonidine reduced sleep bruxism frequency (by 61%). These findings support the hypothesis of sympathetic activity in sleep bruxism.

The study of increased inflammatory markers in sleep bruxers introduced by our research team has the advantage that, to our knowledge, it is the first research of this type in this field. Michalek-Zrabkowska et al. [34] demonstrated that the bruxism episode index positively correlated with the concentrations of plasma C-reactive protein and fibrinogen and 17-hydroxycorticosteroids in the collected urine samples independently of AHI and BMI. Escalated inflammatory markers are associated with increased risk of diabetes and cardiovascular diseases; thus, this aspect of the research suggests increased CV risk associated with SB.

The reviewed literature was also heterogeneous with respect to SB diagnostic criteria. Five recent studies [30,32,33,34,35] defined RMMA/SB events depending on EMG recordings in line with AASM standards [10]. On the other hand, Nashed et al. [29], Huynh et al. [27,28] and Kato et al. [26] diagnosed SB according to validated research criteria demonstrated by Lavigne et al. [38]. Sjoholm et al. [25] examined and counted episodes of masseter contractions per hour of sleep, whereas Okeson et al. [24] defined a bruxing event as “activity of the masseter muscle exceeding 40% of the maximum clench of the muscle and lasting for 2 s or longer”. One study used different SB determination methods such as the original evaluating system by Yoshimi et al. [39] based on the criteria of EMG threshold level and event duration.

## 5. Discussion

Our systematic review has addressed the issue of the cardiovascular implications of sleep bruxism; however, it also reveals a number of gaps and shortcomings. The findings of most studies are still initial and insufficient. An apparent limitation of most studies was small sample size (see Table 1) and variability among subjects, which reduce the significance level. As has been demonstrated in Table 2, according to the GRADE system for assessing the quality of evidence, the majority of studies were low quality due to risk of biases and imprecision. Future studies should aim to replicate results in a larger, randomly assigned population.

Sleep bruxism is a common sleep-related behavior, but it is still insufficiently explored and underdiagnosed [40]. SB was previously linked with many consequences for oral and overall health as primary and secondary factor to different disorders, e.g., sleep disordered breathing, insomnia, headache, temporomandibular disorders or mental dysfunctions as increased anxiety/risk of depression [41]. From the review above, key findings reveal an association between sleep bruxism and increased sympathetic tone, thus increasing blood pressure fluctuations and heart rate variability. They are important cofactors associated with potential cardiovascular consequences of sleep bruxism for general health. The possibility of CV complications warrants changing the current clinical approach in this field.

On the basis of the reviewed articles, we conclude that autonomic function in sleep bruxers is disturbed, introducing a whole range of adverse effects. According to the literature regarding sympathetic hyperreactivity, all three studies confirmed an association between sleep bruxism and sympathovagal balance in relatively young and healthy subjects [25,27,31]. Three studies focused on temporal coincidence between heart rate changes and SB events [24,26,33]. Despite the heterogeneity of revised papers in terms of methodology and study population, the relationship between sleep bruxism and autonomic dysfunction was observed.

Sympathetic dominance in sleep bruxers was also investigated by using two sympatholytic medications—propranolol and clonidine. Clonidine affected sympathetic tone influencing HRV preceding SB onset. On the other hand, propranolol did not affect HRV related with SB events. A broad explanation is that propranolol and clonidine have various properties in terms of pharmacokinetics and pharmacodynamics; clonidine is considered as a more potent medication in sympathetic modulation, due to its impact on alfa-2 adrenergic and imidazoline 1 receptors in the central nervous system [42]. Moreover, clonidine was also linked with silencing modulation in dopaminergic pathways. Importantly, individual response variability in the study group with clonidine was observed [28].

The broad implication of autonomic nervous system stimulation is vasoconstriction. Increased risk of hypertension was previously linked with insomnia, sleep-related breathing disorders and restless leg syndrome [43]. Multiple control systems are involved in blood pressure regulation, but neurogenic dysfunction is considered a crucial component of blood pressure fluctuations in sleep bruxism. Blood pressure dynamics depends on circadian modulation with the influence of environmental, autonomic, emotional and physical factors. As has been established, blood pressure variability is linked with increased risk of organ damage and cardiovascular mortality. It is an important equivalent of hypertension and therefore, overall cardiovascular risk [44]. We identified four studies contributing to blood pressure: the study by Nashed et al. [29] confirmed an association between BP surges and SB events, the study by Michalek-Zrabkowska et al. [35] demonstrated the association between blood pressure variability during nighttime and sleep bruxism intensity in normotensive patients, whereas two studies by Martynowicz et al. [30,32] evaluated the statistically significant relationship between hypertension and sleep bruxism intensity. Future research should further develop and confirm these initial findings.

The majority of prior studies have focused on sympathetic hyperactivity and the cardiovascular complications of sleep bruxism. A recent study by Michalek-Zrabkowska et al. [34] concluded that sleep bruxers may be at increased CV risk due to hormonal disturbances and inflammatory process. As far as we know, no previous research has investigated the relationship between SB and inflammatory response; however, the existing literature confirms the harmful effect of inflammatory mediators to cardiovascular system structures [45,46].

The reviewed studies have varied in terms of study design and methodology. In most studies, sleep bruxism diagnosis was based on polysomnography, the gold standard for SB diagnosis [47]; however, type I polysomnography with video and audio recordings was performed only in five studies [26,29,32,34,35]. Nonetheless, the next three studies consisted of PSGs supplemented with video recordings [24,31,33]. The risk of overestimation related to absence of audio/video recordings was raised by Carra et al. [48] and evaluated up to 25%. The risk of low specificity related to SB event overscoring is increased in level II and III polysomnographic systems and level I PSG without camera recordings. In consonance with Table 1, 3 out of 12 publications were based on the II or III study type according to AASM [27,28,30], but 2 of them had supplementary audio and video recordings [27,28], whereas 1 polysomnographic study was poor in terms of video and audio data [25].

Despite having a number of limitations, these findings provide a potential mechanism for the cardiovascular implications of sleep bruxism. Increased sympathetic activity can be considered as a basic pathomechanism in SB, likely leading to dysfunction of the cardiovascular system. As has previously been shown, sleep bruxism events are associated with microarousals and increase in sympathetic tone [49]. Moreover, Kato et al. [50] demonstrated that experimentally induced arousals were also related with sleep bruxism episodes. Cardiac-autonomic activation influences cortical pathways and induces microarousal and increase in heart rate with subsequent RMMA episode in the end. Moreover, SB events are frequently accompanied by respiratory events. Alternatively, increased sympathetic tone related to SB event could simply be a part of physiological reaction to arousal without an impact on general health.

Although the concept of the genetic heritability of primary sleep bruxism has been discussed by a great number of authors in the literature [5], many questions on the genetic basis of sleep bruxism have remained unanswered for years. Wieckiewicz et al. [3] investigated the possible genetic contribution to the etiology of sleep bruxism and its relationship with obstructive sleep apnea. Taking into consideration theories on serotonin and dopamine pathways’ roles in SB and OSA, researchers investigated the association of selected single-nucleotide polymorphisms (SNPs) within serotonin and dopamine genes. Together, the findings confirmed a possible genetic predisposition to sleep bruxism and its relationship with obstructive sleep apnea [3]. The broad implication of the aforementioned study is that sleep bruxism may be considered an important predisposing factor for severe health conditions such as obstructive sleep apnea. Hence, sympathetic hyperactivity and oxidative stress cause cardiovascular diseases and death. Sleep disturbances accompanied by the chronic inflammatory process increase cardiovascular risk. The proposed cause and effect relationship is presented in Figure 2.

Overall, on the basis of the results of the current review, we speculate that environmental factors and genetic vulnerability to stress and anxiety may constitute the origin of the vicious cycle. Subsequently, the central nervous system is activated in response to chronic stress, affecting neurotransmission and hormonal regulation. Increased autonomic nervous system tone influences sleep architecture, leading to arousals and subsequent RMMA events. Sustained dysregulation and sleep reduction promote the production of free radicals, sympathetic activity, and subsequently, endothelial dysfunction and its consequences: heart rate variability, blood pressure fluctuations and metabolic disturbances. These may constitute risk factors for cardiovascular and metabolic diseases in the future: cardiac arrythmias, hypertension, coronary artery disease, insulin resistance or diabetes. The potential effect on general health condition may also include risk of obesity and metabolic syndrome, so-called civilization diseases. This component of a positive feedback loop accelerates CV disease development. Taking into consideration this domino effect, this is an interesting topic for new multidisciplinary approaches.

According to the PICO question, the broadly translated findings of the current review indicate that sleep bruxism can be considered as a predisposing condition to increased cardiovascular risk. Ideally, these findings should be replicated in a study investigating cardiovascular risk with validated scoring systems, e.g., the SCORE scale or The Framingham Risk Score.

A challenging problem which arises in the domain of sleep bruxism is its treatment, due to its multifactorial etiology and insufficiently determined consequences. A large number of alternative approaches have been developed over the last two decades to manage sleep bruxism: physio- and psychotherapy [51], oral appliances [52,53], CPAP [54], sleep hygiene [55], injections of botulinum toxin or medicaments [6]. However, we acknowledge that there are still considerable discussions among researchers about possible therapeutic paths in sleep bruxism. Several studies have been addressed on the clinical effectiveness of pharmacotherapy, some focusing on beta-blockers or dopamine agents, others on antidepressants. As we have presented in the current review, Huynh et al. [28] evaluated the impact of two sympatholytic medications: propranolol, a nonselective adrenergic β-blocker and clonidine, a selective α2-agonist, indicating that only clonidine influenced sympathetic tone and reduced the sleep bruxism index. Nonetheless, small sample size and morning symptomatic hypotension were apparent and serious study limitations. A recent study by Wieckiewicz et al. [56] concluded that opipramol, an atypical tricyclic antidepressant, reduced the amount of SB events (decreased BEI) in 78.95% of severe bruxers. Although the results appear promising, they suffer from study limitations, e.g., lack of control group, small study sample and lack of longitudinal observation to estimate the side-effects of the applied pharmacotherapy. The possibility of severe cardiovascular implications of sleep bruxism warrants further investigation of targeted therapies.

Collectively, a number of proposed recommendations for future research and clinical approaches are given in Table 3.

Overall, the results of the current systematic review strongly suggest an association between sleep bruxism and its cardiovascular implications. Evidence synthesis demonstrated a reliable pathomechanism linking SB and sympathetic hyperactivity, and subsequently, potential increased CV risk in sleep bruxers. The main concern about the findings of the present research were the limitations of the presented studies. Most studies have relied on a small sample size and observational design. Moreover, the heterogeneity of the discussed papers in terms of studied population, study design, polysomnographic study type, SB diagnostic criteria and, last but not least, aim of the study foreclose a more profound synthesis of investigated issue. Although the results of the aforementioned publications are inconclusive, further attempts could prove quite beneficial not only for oral, but also the general health of sleep bruxers.

## Figures and Tables

**Figure 1 jcm-10-02245-f001:**
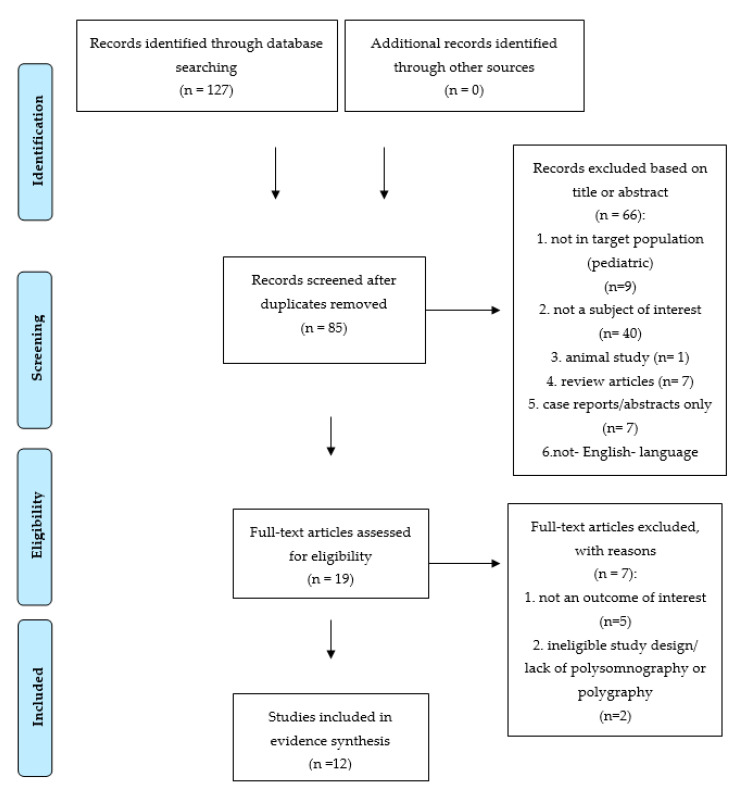
PRISMA (Preferred Reporting Items for Systematic Reviews and Meta-Analyses) flow chart of the review protocol.

**Figure 2 jcm-10-02245-f002:**
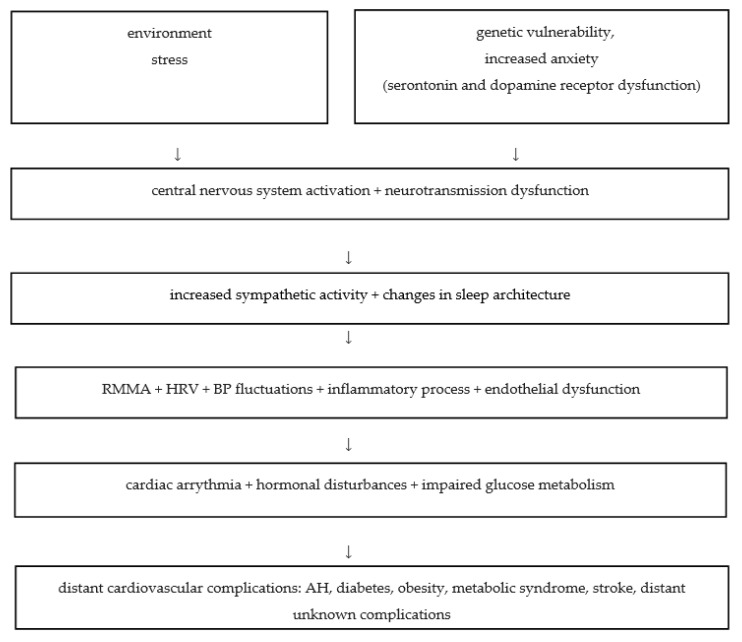
A possible cause-and-effect diagram of local and overall cardiovascular consequences of sleep bruxism. RMMA, rhythmic masticatory muscles activity; HRV, heart rate variability; BP, blood pressure; AH, arterial hypertension.

**Table 1 jcm-10-02245-t001:** The main characteristics of the revised studies.

Author and Year	Study Design	Study Population (Cases/Controls)	Compared Groups	Study Type *	Aim of the Study	Comments
Okeson et al., 1994 [24]	observational	20	none	I	to collect normative data on “nocturnal bruxing events”	none
Sjoholm et al., 1995 [25]	observational	11	none	I	to examine autonomic function in “nocturnal toothgrinders”	none
Kato et al., 2001 [26]	case–control	20 (10/10)	SB patients vs. normal	I	to evaluate association among autonomic-cardiac, cortical, and jaw muscle activities, to determine a sequence of events associated with RMMA and bruxism during sleep	none
Huynh et al., 2006 [27]	case–control	60 (40/20, see comments)	moderate to high SB vs. low SB vs. control subjects	III	to assess: (1) the distribution of RMMA referring to sleep stage and sleep cycles; (2) the time correlation between RMMA and microarousals referring to SWA dynamics over sleep cycles in three study groups; (3) the time correlation between SB activity and autonomic cardiac activity	2 study groups: moderate to severe SB *n* = 20 patients; low SB *n* = 20 patients; control group *n* = 20 subjects
Huynh et al., 2006 [28]	randomized control trial	25 (see comments)	propranolol vs. clonidine	III	to examine whether (1) propranolol or clonidine may reduce the occurrence of SB; (2) may prevent the rise in autonomic sympathetic activity preceding the onset of SB	Study with propranolol n = 10 subjects; study with clonidine n = 16 subjects; 1 patient participated in both studies
Nashed et al., 2012 [29]	case–control	24 (10/9 active subjects)	SB patients vs. normal	I	to determine association between BP surges and SB events in SB subjects in relation to arousals and/or body movements	5 of the 14 recordings in control group had technical difficulties; *n* = 9
Martynowicz et al., 2018 [30]	case–control	70 (35/35)	hypertensives vs. normal	III	to examine SB severity in hypertensives compared to normotensives	none
Nukazawa et al., 2018 [31]	case–control	11	none	I	to investigate the relationship between SB and AN system activity	none
Martynowicz et al., 2019 [32]	observational	87	see comments	I	to assess the relationship between SB intensity and serum renalase concentration	SB patients vs. normal; hypertensives vs. normotensives; selected according to results of the study
Zhong et al., 2020 [33]	case–control	21 (10/11)	SB patients vs. normal	I	to evaluate HRV in relation to: SB types, RMMAs + LMs and isolated LMs in sleep bruxers	none
Michalek-Zrabkowska et al., 2020 [34]	observational	74	none	I	to diagnose sleepiness, hormonal changes and inflammatory markers (CRP, fibrinogen) in SB patients	none
Michalek-Zrabkowska et al., 2021 [35]	observational	65	none	I	to assess the association between ambulatory blood pressure measurements and SB intensity in normotensive individuals	none

* study type according to AASM criteria [36]; SB, sleep bruxism; SWA, slow wave activity; RMMA, rhythmic masticatory activity; BP, blood pressure; AN, autonomic nervous system; LM, limb movement; CRP, C-reactive protein.

**Table 2 jcm-10-02245-t002:** Quality of the evidence of the included studies.

Outcome Significance	Author and Year	Quality of the Evidence
		(GRADE System)
non-significant	Okeson et al., 1994 [24]	+--- very low due to high risk of bias, imprecision
non-significant	Sjoholm et al., 1995 [25]	+--- very low due to high risk of bias, imprecision
significant	Kato et al., 2001 [26]	++-- low due to risk of bias, imprecision
significant	Huynh et al., 2006 [27]	+++- moderate due to large effect
significant	Huynh et al., 2006 [28]	++-- low due to risk of bias, imprecision
significant	Nashed et al., 2012 [29]	++-- low due to risk of bias, imprecision
significant	Martynowicz et al., 2018 [30]	+++- moderate due to large effect
significant	Nukazawa et al., 2018 [31]	++-- low due to risk of bias, imprecision
significant	Martynowicz et al., 2019 [32]	+++- moderate due to large effect
significant	Zhong et al., 2020 [33]	++-- low due to risk of bias, imprecision
significant	Michalek-Zrabkowska et al., 2020 [34]	+++- moderate due to large effect
significant	Michalek-Zrabkowska et al., 2021 [35]	+++- moderate due to large effect

GRADE scores: +++- (moderate), ++-- (low), +--- (very low).

**Table 3 jcm-10-02245-t003:** Practice points and research agenda proposed by authors.

No.	Practice Points	Future Research
1	Prior publications have suggested the role of increased sympathetic activity in sleep bruxism and its link with cardiovascular implications.	Further research is certainly required to determine distant effects of sleep bruxism on cardiovascular system and general health.
2	Studies on the relationship between SB and most common sleep-related disorders are well documented; for example, the association between sleep bruxism and obstructive sleep apnea or insomnia.	Future studies should aim to investigate the association between SB and civilization diseases: obesity, diabetes and hypertension.
3	A number of authors have documented CV implications in SB subjects as heart rate variability or blood pressure fluctuations.	Future studies should aim to replicate results in a larger population.
4	The existing literature suggests an association between SB intensity and increased CV risk.	Therefore, future research should be conducted in respect to standards and guidelines for clinical trials to obtain statistical power.
5	A more comprehensive description of increased sympathetic tone in SB assumes that genetic vulnerability and exposure to stress induces a cascade of reactions in the central and autonomic nervous systems with broad implications for overall health.	The sleep and bruxism data should be investigated with level 1 polysomnography, supplemented with audio and video recordings to avoid overestimation of bruxism events.
6	Cardiovascular implications of sleep bruxism have rarely been studied directly.	We propose that SB events should be evaluated according to the criteria of the American Academy of Sleep Medicine.
7	Previous studies on this subject cannot be considered as conclusive because of lack in statistical power and limitations.	The possibility of the cardiovascular implications of sleep bruxism warrants further longitudinal investigation.

SB, sleep bruxism; CV, cardiovascular.

## Data Availability

The data presented in this study are openly available in databases: Scopus, PubMed and Embase.

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
