# Peer review of "Cardiovascular Implications of Sleep Bruxism—A Systematic Review with Narrative Summary and Future Perspectives"

_jcm, 2021, doi:10.3390/jcm10112245_

Round 1

Reviewer 1 Report

Comments on ”cardiovascular implications of sleep bruxism…”

Introduction: In the first paragraph you state that sleep bruxism is not a sleep disorder. But in the second paragraph you give information about associations between sleep-related disorders and cardiovascular complications.

As sleep bruxism has not been classified as a sleep disorder, please, rewrite the second section regarding sleep disturbances and your aim to search for such effects also from sleep bruxism. OSA has a much different background and effects compared to sleep bruxism.

Already in the introduction you should mention that there are different diagnostic criteria for sleep bruxism.

Results: In the third paragraph you “confirm the association between SB and cardiovascular disturbances”. Please, replace the word disturbances with response or influence. Try not to speculate about the findings of the review under the Result.

Line 198 to 201 should be moved to conclusion at the end of discussion.

Line 202, the evaluation with GRADE system should be described under methods and with a reference.

Discussion: The discussion is too long, with very long sections that has to be divided, and much information or thoughts are given both under results and discussion.

One factor to be discussed is that bruxism is frequent during REM-sleep and arousal, which are associated with cardiovascular response. Increased heart beat and blood pressure also follow physical training and is a physiologic reaction not regarded as harmful in proper amount. The time (frequency and length) of SP must have some significance but is not mentioned regarding any CV risk. Be careful to speculate about the cardio vascular repose from SB as a CV risk.

Start the Discussion with the section placed at line 271.

Move the section starting at line 291 to the Introduction.

Please, try to condense the discussion

In figure 2 change the legend to “A possible cause-and-effect…”.

Line 352 and following you must stress that the possible explanation you give is your thoughts and not evidence from the literature review, or?.

Line 198 - 201 should be moved to the conclusion at the end of discussion.

Author Response

Introduction: In the first paragraph you state that sleep bruxism is not a sleep disorder. But in the second paragraph you give information about associations between sleep-related disorders and cardiovascular complications. As sleep bruxism has not been classified as a sleep disorder, please, rewrite the second section regarding sleep disturbances and your aim to search for such effects also from sleep bruxism. OSA has a much different background and effects compared to sleep bruxism.

Thank you, we hope that reconstruction of introduction will provide better understanding of issue. Although sleep bruxism is considered as sleep behavior rather than sleep disorder, previous studies have emphasized its coexistence with sleep- related disorders, for example obstructive sleep apnea. Most of SB events are preceded by brain and cardio-vascular activity, thus, because of the link between SB and OSA such cardiovascular effects can potentially influence both SB and sleep – related disorders.

Already in the introduction you should mention that there are different diagnostic criteria for sleep bruxism.

We have extended this part of introduction section, thank you for your notice.

Results: In the third paragraph you “confirm the association between SB and cardiovascular disturbances”. Please, replace the word disturbances with response or influence. Try not to speculate about the findings of the review under the Result.

Thank you, we have already revised manuscript including your suggestions.

Line 198 to 201 should be moved to conclusion at the end of discussion.

We have removed this part from Results.

Line 202, the evaluation with GRADE system should be described under methods and with a reference.
Thank you, we have already added the reference and we have changed it according to your suggestions.

Discussion: The discussion is too long, with very long sections that has to be divided, and much information or thoughts are given both under results and discussion.

One factor to be discussed is that bruxism is frequent during REM-sleep and arousal, which are associated with cardiovascular response. Increased heart beat and blood pressure also follow physical training and is a physiologic reaction not regarded as harmful in proper amount. The time (frequency and length) of SP must have some significance but is not mentioned regarding any CV risk. Be careful to speculate about the cardio vascular repose from SB as a CV risk.

This is very important and valuable comment, thank you. Results of current study provided evidence for probable implications of SB for cardiovascular system. On the other hand, increased sympathetic tone related to SB event could simply be a part of physiological reaction to arousal without impact on general health. As you have suggested, we discussed it.

Start the Discussion with the section placed at line 271. Move the section starting at line 291 to the Introduction.

Thank you, we have already revised manuscript including your suggestions.

Please, try to condense the discussion.

Thank you for your advice, we revised and reduced the discussion.

In figure 2 change the legend to “A possible cause-and-effect…”.

We have already changed it according to Reviewer’s opinion.

Line 352 and following you must stress that the possible explanation you give is your thoughts and not evidence from the literature review, or?.

Thank you, we have emphasized that this section of discussion is based on our speculations according to results of current work.

Reviewer 2 Report

Thank you for sharing your manuscript with me.

It is very well-designed, following the publication rules, and answers relevant questions.

some minor points:

  1. Tables are overloaded. some spacing will improve the overview.
  2. Table 1-study type according to AASM criteria- please provide a reference.
  3. The use of the PRISMA system is appreciated. However, the use of PRISMA 2020 will be more updated (the PRISMA 2020 statement: an updated guideline for reporting systemic reviews BMJ2021;372:n71 ).
  4. Reference 20 is missing the year of publication.

Major points:

  1. Due to heterogeneity and variety in terms of study design and methodology only comparison and narrative summary were performed. That is a very important issue that needs to be added to the title.
  2. Quality assessment of the reviewed papers (external and internal validity ) is missing. Add or Explain.

 I do have a few concerns (described in comments for authors) and I believe mainly minor changes are needed.

with the publication rules, and answers relevant questions.

some minor points:

  1. Tables are overloaded. some spacing will improve the overview.
  2. Table 1-study type according to AASM criteria- please provide a reference.
  3. The use of the PRISMA system is appreciated. However, the use of PRISMA 2020 will be more updated (the PRISMA 2020 statement: an updated guideline for reporting systemic reviews BMJ2021;372:n71 ).
  4. Referance 20 is missing the year of publication.

    Major points:

    1. Due to heterogeneity and variety in terms of study design and methodology only comparison and narrative summary were performed. That is a very important issue that needs to be added to the title.
    2. Quality assessment of the reviewed papers (external and internal validity ) is missing. Add or Explain.
  5.  

Author Response

Tables are overloaded. some spacing will improve the overview.

We have already changed it according to Reviewer’s opinion.

Table 1-study type according to AASM criteria- please provide a reference.

Thank you, we have already added the reference.

The use of the PRISMA system is appreciated. However, the use of PRISMA 2020 will be more updated (the PRISMA 2020 statement: an updated guideline for reporting systemic reviews BMJ2021;372:n71).

Reference 20 is missing the year of publication.

Thank you, we have already added the up- to- date reference and revised all of them.

Major points:

Due to heterogeneity and variety in terms of study design and methodology only comparison and narrative summary were performed. That is a very important issue that needs to be added to the title.

We have extended the title, thank you for your notice.

Quality assessment of the reviewed papers (external and internal validity ) is missing. Add or Explain.

Quality of the evidence of the included studies was made according to GRADE system- scores and explanations were included in Table 2. GRADE criteria can be used as indicators of confidence in systematic reviews: internal validity of a study, consistency, publication bias.

 I do have a few concerns (described in comments for authors) and I believe mainly minor changes are needed.